# Leriche Syndrome Misdiagnosed as Complex Regional Pain Syndrome in a Patient with Neuropathic Pain Caused by a Chip Fracture: A Case Report

**DOI:** 10.3390/medicina57050486

**Published:** 2021-05-12

**Authors:** Byeong-Cheol Lee, Dae-Seok Oh, Hyun-Seong Lee, Se-Hun Kim, Jae-Hong Park, Ki-Hwa Lee, Hyo-Joong Kim, Ji-Hyun Yang, Sang-Eun Lee

**Affiliations:** Department of Anesthesiology and Pain Medicine, Haeundae Paik Hospital of Inje University, 875 Haeundae-ro, Haeundae-gu, Busan 48108, Korea; ironism00@gmail.com (B.-C.L.); yivangin@naver.com (D.-S.O.); leggi-ero24@gmail.com (H.-S.L.); anesehunkim@outlook.kr (S.-H.K.); H00150@paik.ac.kr (J.-H.P.); tedy333@paik.ac.kr (K.-H.L.); H00386@paik.ac.kr (H.-J.K.); yang557984@gmail.com (J.-H.Y.)

**Keywords:** claudication, complex regional pain syndrome, Leriche syndrome, neuropathic pain, peripheral arterial disease

## Abstract

Introduction: Leriche syndrome is an aortoiliac occlusive disease caused by atherosclerotic occlusion. We report a case of Leriche syndrome with a fracture that was suspected as complex regional pain syndrome (CRPS), as the post-traumatic pain gradually worsened in the form of excruciating neuropathic pain. Case Report: A 52-year-old woman with a history of hypertension was referred to the Department of Pain Medicine from a local orthopedic clinic because of suspected CRPS for excruciating neuropathic pain for one month. She complained of gait dysfunction and severe pain in the right foot following an incident of trauma with the right first toe. The average pain intensity assessed using the visual analog scale (VAS) was 90 (0: no pain, 100: the worst pain imaginable), and the neuropathic pain was evident as a score of 6/10 on Douleur neuropathique 4. Allodynia, hyperalgesia, blue discoloration of the skin, asymmetric temperature change (1.38 °C), and edematous soft tissue changes were observed. Ultrasonography showed a chip fracture in the first distal phalanx of the right first toe. The diagnosis was most probably CRPS type I according to the Budapest research criteria for CRPS. However, multiple pain management techniques were insufficient in controlling the symptoms. A month and a half later, an ankle-brachial index score of less than 0.4 suggested severe peripheral artery disease. Computed tomography angiography showed total occlusion between the infrarenal abdominal aorta and the bilateral common iliac arteries. Therefore, she underwent aortic-bifemoral bypass surgery with a diagnosis of Leriche syndrome. Three months after the surgery, the average pain intensity was graded as 10 on the VAS (0–100), the color of the skin of the right first toe improved and no gait dysfunction was observed. Conclusion: A chip fracture in a region with insufficient blood flow could manifest as excruciating neuropathic pain in Leriche syndrome.

## 1. Introduction

Leriche syndrome, an aortoiliac occlusive disease, is the thrombotic obliteration of the terminal part of the abdominal aorta, which may be asymptomatic or may present as intermittent claudication, diminished femoral pulse, lower-extremity pallor and/or impotence, similar to peripheral arterial disease (PAD) [1,2,3]. PAD is especially difficult to diagnose, and Leriche syndrome is often misdiagnosed.

PAD gradually leads to vascular insufficiency to cause ischemic pain from intermittent claudication or chronic limb-threatening ischemia, which manifests as rest pain and ulceration depending on the location and severity of atherosclerotic occlusion in the lower extremities [4,5]. Generally, patients with intermittent claudication present with nociceptive pain, while those with critical limb ischemia predominantly experience neuropathic pain [5,6,7,8].

The diagnoses of neuropathic pain, and of the causative lesions and/or diseases in PAD, are difficult [9,10]. Peripheral neuropathic pain not only manifests gradually in chronic ischemia, but is also inversely related to the degree of peripheral blood flow [11]. We report a case of Leriche syndrome with chip fracture of the first toe, which triggered acute excruciating neuropathic pain in the right foot. Acute ischemic resting pain of the limb is often misdiagnosed as complex regional pain syndrome (CRPS) in the absence of an embolic or thrombotic arterial occlusion.

## 2. Case Report

A 52-year-old woman was referred to the Department of Pain Medicine from a local orthopedic clinic because of suspected CRPS for excruciating neuropathic pain in her right first toe. She was 160 cm tall and weighed 55 kg. She was taking antihypertensive medications and had undergone a surgery for varicose veins in both legs five years ago.

Six weeks previously, she bumped into something with her right first toe. She complained of gradually worsening pain in her right foot after the incident, and the skin over the region had turned cold and purple due to edema over time (Figure 1).

She complained of gait dysfunction and severe pain in the right foot, especially in the first toe, of burning, tingling, prickling, stabbing, heavy and of aching type. Pain gradually spread to the right ankle, calf, and thigh, and she reported the average pain intensity assessed using the visual analog scale (VAS) as 90 (0: no pain, 100: the worst pain imaginable). She presented with numbness, pinprick hyperalgesia, pressure hyperalgesia, static mechanical allodynia and dynamic mechanical allodynia in her right foot. Neuropathic pain was dominant in her foot with a score of 6/10 on Douleur neuropathique 4 (DN4) and 36/38 on the painDETECT questionnaire (PDQ).

Blood tests and radiographs of the right foot did not show any abnormal findings. Ultrasonography showed a chip fracture in the first distal phalanx of the right first toe (Figure 2). Therefore, she was diagnosed with post-traumatic neuropathic pain associated with a toe fracture. However, the pain worsened over time despite multimodal pain management. Additional tests such as three-phase bone scan, digital infrared thermal imaging, quantitative sudomotor axon reflex test, electromyography and a nerve conduction study were performed to exclude CRPS.

The three-phase bone scan showed decreased uptake in the right foot during the blood pooling phase, and mildly increased uptake in the right first toe during the osseous phase, which were unusual findings suggestive of CRPS (Figure 3).

Digital infrared thermal imaging showed that the temperature of the right foot was lesser than that of the left, and the difference in temperature was approximately 1.38 °C. No sweating dysfunction or peripheral neuropathy in the lower extremity were noted in the quantitative sudomotor axon reflex test and electromyography and nerve conduction study. Clinically, the patient could be diagnosed with subacute CRPS type I according to the Budapest research criteria for CRPS.

Although various pain management techniques, such as the use of pregabalin, amitriptyline, tapentadol, morphine, ketamine, lidocaine, epidural blocks and sciatic nerve blocks, were employed in accordance with the diagnosis of CRPS type I, she only had temporary pain relief, and her gait function worsened. The pain continued to worsen and her right toe turned dark purple in color, making it impossible to maintain her daily activities.

The dark purple skin color was suspected to be caused by gangrene as it was accompanied by weak femoral pulses on physical examination. The ankle-brachial index (ABI) score was 0.4, suggestive of severe PAD. Lower-extremity computed tomography (CT) angiography, performed in anticipation of PAD, revealed total occlusion between the infrarenal abdominal aorta and the bilateral common iliac arteries (Figure 4).

As a result, the patient was diagnosed with Leriche syndrome. She underwent an aortic-bifemoral bypass surgery, following which the patient required cardiopulmonary rehabilitation for two months and showed no other complications.

One month later, her neuropathic pain improved with a DN4 score of 1/10 and PDQ score of 5/28, and average pain intensity improved to 30 on the VAS (0–100). Her gait and the color of the skin over the toe also improved. Five months postoperatively, she was only prescribed pregabalin 75 mg/day, with a normal color of the skin over the toe and mild pain intensity with a VAS score of 10 (0–100) in the absence of gait-evoked pain.

## 3. Discussion

Leriche syndrome reduces blood flow to the pelvic region and lower extremities similar to PAD [3]. Leriche syndrome can be asymptomatic or may present as intermittent claudication after physical exercise, which resolves with rest [1,2,12,13]. Therefore, PAD cannot be diagnosed until the presentation of intermittent claudication.

In this case, a fracture complicated by ischemic tissue damage in an asymptomatic patient with Leriche syndrome caused extremely severe neuropathic pain without any evidence of thromboembolism. As bone mineral density decreases with age, the risk of fracture increases [14]. A fracture may diminish perfusion concurrent with the increased metabolic demands of repair. Hence, minor damage may cause a fracture, worsen the hypoxia and acidosis near the fracture site and provoke ischemic resting neuropathic pain in Leriche syndrome.

Ischemic neuropathy is often observed with progressive chronic ischemia of the peripheral nerves [7,15]. Several studies have also shown that the incidence and severity of peripheral neuropathy are associated with the severity of ischemia. Chronic ischemia causes activity-dependent nociceptive stimuli and inflammation to manifest as intermittent claudication. Moreover, severe limb ischemia in progressive PAD, accompanied by critical limb ischemia with resting pain, presents predominantly with neuropathic pain [5,6,7,8,16].

In this case, few or practically no clinical features of ischemic neuropathy presented before the traumatic injury. However, neuropathic pain was distinctly evident in the right foot afterward, with signs of numbness, hyperalgesia and allodynia, as shown by a 6/10 score on DN4 and a 36/38 score on PDQ. There was no evidence of neuropathic pain secondary to peripheral neuropathy except Leriche disease, although DN4 and PDQ scores were suggestive of neuropathic pain. DN4 and PDQ were utilized to distinguish neuropathic pain from nociceptive pain, as these screening tools evaluate the verbal descriptors and pain qualities.

Both nociceptive and neuropathic pain underlay the patient’s symptoms and could have been the result of ischemia in non-neural tissues such as the skin and muscle. This patient had edematous and purple-colored skin on the first right toe with severe neuropathic pain owing to gangrene. Gangrene is a type of tissue death caused by a lack of blood supply, which commonly occurs in the toes [17,18]. In this case, the blue and purple discoloration of the skin, soft tissue swelling, numbness and severe pain were associated with gangrene. A small traumatic injury resulted in gangrene due to insufficient blood perfusion in the first toe, with the loss of compensatory function in Leriche syndrome.

Bone has extensive vascularization. The vascular network within and around the bone is disrupted in bone fractures, accompanied by damage to the bone marrow and surrounding soft tissue [19]. As a result, decreased perfusion concurrent with the increased metabolic demands for union leads to hypoxia near the fracture site [20,21]. Hypoxia can worsen and prevent fracture healing under chronic ischemic condition associated with Leriche syndrome. Therefore, this patient’s neuropathic pain could have been associated with the unhealed fracture complicated by exacerbation of chronic limb ischemia [6,16]. The excruciating neuropathic pain subsided after aorto-bifemoral bypass surgery, supporting the hypothesis that Leriche syndrome was the leading cause of post-traumatic neuropathic pain. In other words, even in an asymptomatic patient with Leriche syndrome, fracture can progress to acute neuropathic pain as critical limb ischemia.

The excruciating neuropathic pain in our patient after a chip fracture in the first toe was misdiagnosed as CRPS. The mechanism of CRPS has not been fully elucidated, rendering it difficult to diagnose. CRPS is characterized by persistent regional pain, abnormal sensory and motor function, vasomotor abnormalities and trophic changes [22,23]. In this case, the patient presented with allodynia, hyperalgesia, skin color changes, edema, asymmetrical temperature changes and muscle weakness in the foot. Therefore, the symptoms and signs of this patient met the Budapest diagnostic criteria based on the following four categories: sensory, vasomotor, sudomotor and motor/trophic [22,23]. In addition, the three-phase bone scan supported the diagnosis of CRPS, which was decreased uptake in the right foot during the blood pooling phase [24].

Several reasons could be attributed to the long duration of three months for the diagnosis of Leriche syndrome despite the presence of neuropathic pain. First, the arteries of the lower extremity were gradually obstructed from the infrarenal abdominal aorta. Therefore, collateral blood vessels could have developed in both lower extremities. This syndrome progresses slowly over a long period. The patient was not aware of the ischemic symptoms until the trauma precipitated neuropathic pain. Second, she was not a typical high-risk PAD patient. Hence, the ABI test was performed after six weeks following the failure of multiple treatments. Third, the course of neuropathic pain was not typical in this patient. The mechanism of neuropathic pain is still not fully understood. According to the Budapest diagnostic criteria, there is no definitive diagnostic test for neuropathic pain, which could have led to the misdiagnosis of CRPS based on the clinical assessment of the patient’s symptoms and signs.

Some reports have shown the misdiagnoses of Leriche syndrome as other diseases. Yoon et al. reported that a patient complaining of pain in the right leg with claudication was initially suspected to have spinal stenosis but was diagnosed with sciatic neuropathy due to Leriche syndrome [13]. Groth et al. suggested that cauda equina syndrome was associated with Leriche syndrome in a patient with symptoms of weakness in the limb and dysfunction of the bladder [12]. Lai et al. reported a case of acute paraplegia due to Leriche syndrome [25]. As such, when evaluating patients with peripheral neuropathy, it is necessary to consider ischemic neuropathy associated with PAD such as Leriche syndrome.

## 4. Conclusions

This case is the first report of Leriche syndrome presenting as acute extreme neuropathic pain in the absence of thromboembolism. When ischemic tissue damage caused by fracture triggers severe neuropathic pain in a patient with unknown asymptomatic Leriche syndrome, it may lead to CRPS misdiagnosis. We assumed that the patient had neuropathic pain in the lower extremities. In such cases, comprehensive history and physical examination should focus on neurological and vascular function, including the ABI test for accurate diagnosis of peripheral neuropathy and excluding Leriche syndrome as a PAD.

## Figures and Tables

**Figure 1 medicina-57-00486-f001:**
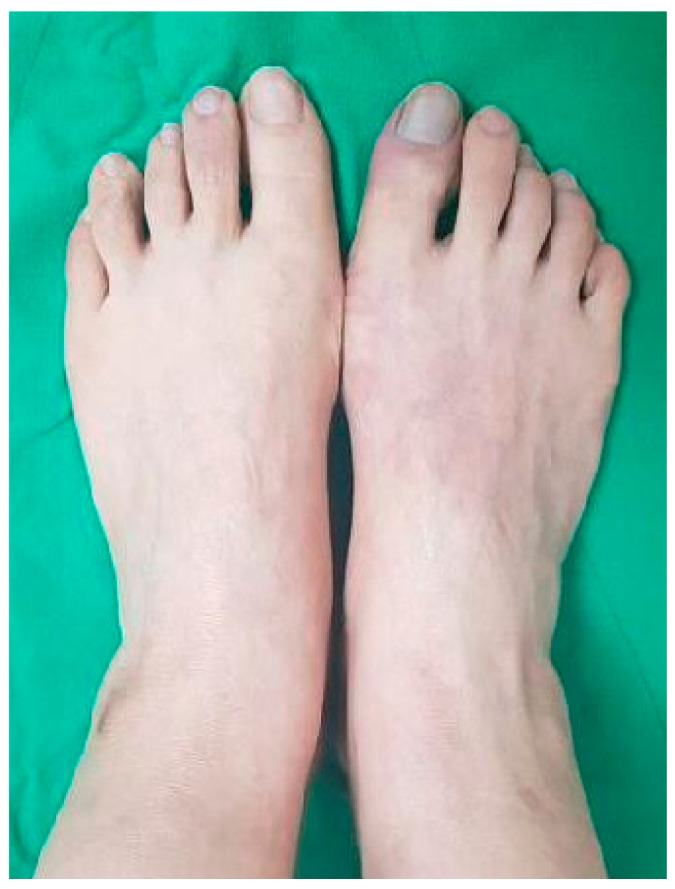
Purple discoloration of the right first toe.

**Figure 2 medicina-57-00486-f002:**
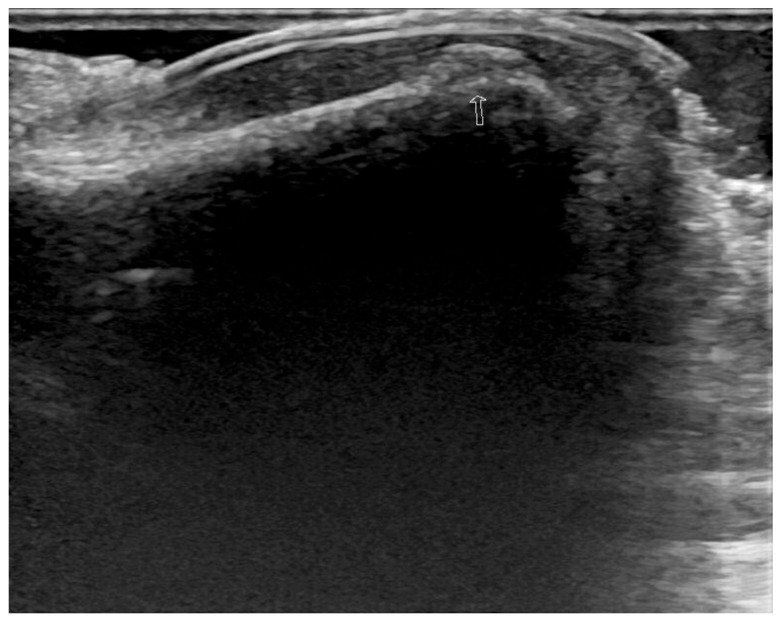
Unilateral foot ultrasonography: Cortical disruption at the dorsal aspect of right first distal phalanx (arrow).

**Figure 3 medicina-57-00486-f003:**
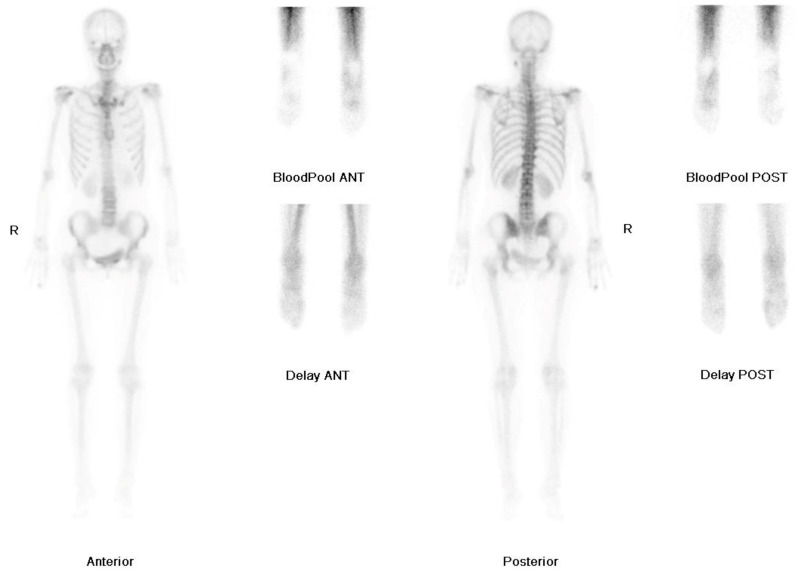
Three-phase bone scan: Decreased uptake in the right foot during the blood pooling phase and mildly increased uptake in the right first toe during the osseous phase.

**Figure 4 medicina-57-00486-f004:**
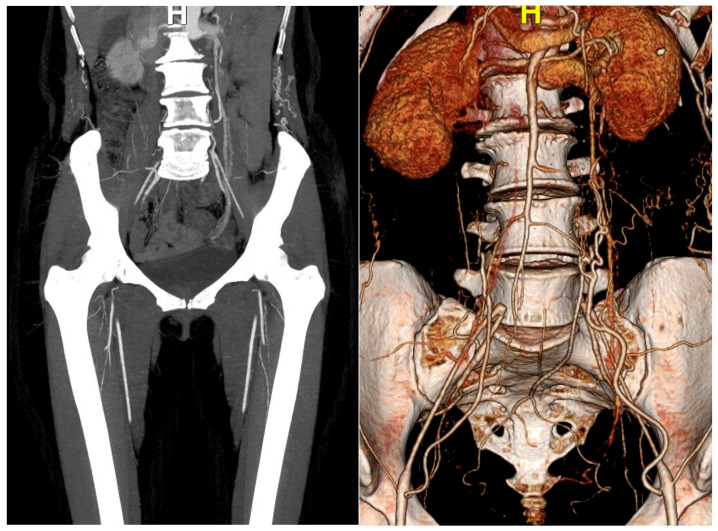
Lower-extremity computed tomography angiography: Total occlusion between the infrarenal abdominal aorta and the bilateral common iliac arteries.

## Data Availability

The data presented in this study are available on request from the corresponding author.

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
