# Peer review of "Leriche Syndrome Misdiagnosed as Complex Regional Pain Syndrome in a Patient with Neuropathic Pain Caused by a Chip Fracture: A Case Report"

_medicina, 2021, doi:10.3390/medicina57050486_

Round 1
Reviewer 1 Report
I greatly enjoyed reading the manuscript titled "Leriche Syndrome Misdiagnosed as Complex Regional Pain Syndrome in a
patient with Neuropathic Pain Caused by a Chip Fracture: A Case Report." I have some comments and recommendations.
Case Report
- Line 71: please provide information on sensory signs (i.e. pinprick hyperalgesia, dynamic mechanical allodynia, pressure hyperalgesia)
- Line 71 and 110: it would be of interest to have the NPSI score too.
- Line 77 and 91: ‘nerve conduction velocity’ please change to ‘nerve conduction study’
Discussion
- Line 125-132: ‘Neuropathic pain in PAD is not easily diagnosed. Ischemic neuropathy is often observed with progressive chronic ischemia of the peripheral nerves [7,12]. Chronic ischemia causes activity-dependent nociceptive stimuli and inflammation to manifest as intermittent claudication Progressive PAD accompanied by critical limb ischemia with resting pain presents predominantly as neuropathic pain [8,13]. Several studies have also shown that the incidence and severity of peripheral neuropathy are associated with the severity of ischemia. Severe limb ischemia may cause neuropathic pain related to nerve dysfunction [5-8,13].’
This should be better explained (i.e. ‘Ischemic neuropathy is often observed with progressive chronic ischemia of the peripheral nerves. Several studies have also shown that the incidence and severity of peripheral neuropathy are associated with the severity of ischemia. Chronic ischemia causes activity-dependent nociceptive stimuli and inflammation to manifest as intermittent claudication. Moreover, severe limb ischemia in progressive PAD accompanied by critical limb ischemia with resting pain, presents predominantly with neuropathic pain.’)
- Line 141: You state: ‘Pain could have been the result of ischemia in non-neural tissues such as the skin and muscle.’
Please make clear to the reader that both nociceptive and neuropathic pain underlie patient’s symptoms.
Conclusions
- Line 191: ‘When neuropathic pain is complicated by ischemic tissue damage caused by fracture in a patient with asymptomatic Leriche syndrome, the rarity of Leriche syndrome may lead to the misdiagnosis of CRPS.’
It can be replaced with: ‘When ischemic tissue damage caused by fracture triggers severe neuropathic pain, in a patient with unknown asymptomatic Leriche syndrome, it may lead to CRPS misdiagnosis.’
Author Response
Thank you for reviewing our manuscript.
Please see the attachment.

Reviewer 2 Report
Dear All,
A case report with a diagnosis of permanent interest.
All the best!
Author Response
Thank you for reviewing our article.
Please see the attachment.

Reviewer 3 Report
A very interesting paper, just mino suggestions.
- In the introduction cite some important paper on neuropathic pain are missing as:
- Management of neuropathic pain: A graph theory-based presentation of literature review. Breast J. 2020 Mar;26(3):581-582. doi: 10.1111/tbj.13622.
- Neuropathic Pain in the Elderly. Diagnostics (Basel). 2021 Mar 30;11(4):613. doi: 10.3390/diagnostics11040613.
About the fracture please cite: Bone mineral density in adults with Down syndrome. Osteoporos Int. 2017. PMID: 28685282, to underline the importance of possible comorbities.
In the discussione the translational value should be underline.
Many thanks
Author Response

(The authors gave the same response as above.)
